# A Cross-Sectional Study of the Gut Microbiota Composition in Moscow Long-Livers

**DOI:** 10.3390/microorganisms8081162

**Published:** 2020-07-30

**Authors:** Daria A. Kashtanova, Nataliya S. Klimenko, Irina D. Strazhesko, Elizaveta V. Starikova, Oksana E. Glushchenko, Denis A. Gudkov, Olga N. Tkacheva

**Affiliations:** 1The Stand-Alone Structural Unit of the Pirogov Russian National Research Medical University the “Russian Clinical Research Center for Gerontology” of the Ministry of Healthcare of the Russian Federation, 129226 Moscow, Russia; istrazhesko@gmail.com (I.D.S.); tkacheva@rgnkc.ru (O.N.T.); 2Centre for Strategic Planning and Management of Biomedical Health Risks of the Federal Medical-Biological Agency, 119435 Moscow, Russia; 3Institute of Gene Biology, Russian Academy of Sciences, 119334 Moscow, Russia; natasha.klmnk@gmail.com; 4Federal Research and Clinical Center of Physical-Chemical Medicine of Federal Medical Biological Agency, 119435 Moscow, Russia; hed.robin@gmail.com (E.V.S.); info@fnkc-fmba.ru (O.E.G.); gudkovdenis@gmail.com (D.A.G.)

**Keywords:** gut microbiota, longevity, aging, systemic low-grade inflammation, long-livers, beneficial bacteria, microbiome

## Abstract

The aim was to assess the gut microbiota of long-livers from Moscow. This study included two groups of patients who signed their consent to participate. The group of long-livers (LL) included 20 participants aged 97–100 years (4 men and 16 women). The second group included 22 participants aged 60–76 years (6 men) without clinical manifestations of chronic diseases (healthy elderly). Gut microbiota was studied by 16S rRNA sequencing. Long-livers underwent a complex geriatric assessment as well as expanded blood biochemistry. Gut microbiota composition in the cohorts was also compared with microbiome in long-livers from Japan and Italy. Russian long-livers’ microbiome contained more beneficial bacteria than healthy elderly including *Ruminococcaceae, Christensenellaceae, Lactobacillaceae* families. Conditional pathogens like *Veillonellaceae, Mogibacteriaceae, Alcaligenaceae, Peptococcaceae, Peptostreptococcaceae* were more abundant in the healthy elderly. Compared with Italian and Japanese microbiome LL, the Russian LL appeared to be more similar to the Italian cohort. *Bifidobacterium/Coprococcus* and *Faecalibacterium/Coprococcus* balances were associated with femoral and carotid intima–media thickness, respectively. *Bifidobacterium/Coriobacteriaceae* balance was assessed with the folic acid level and *Faecalibacterium/Coriobacteriaceae_u* the with Mini Nutritional Assessment score. Long-livers’ microbiome appeared to be unexpectedly balanced. The high representation of beneficial bacteria in long-livers may prevent them from low-grade inflammation and thus protect them from the development of atherosclerosis and other aging-associated conditions.

## 1. Introduction

The planet’s population is steadily aging, and the healthcare system is faced with the task of prolonging the active period of life and finding methods for the effective prevention of age-associated conditions that impose a large burden not only on medicine, but also on the economy as a whole. One of the closely studied and promising protective factors is gut microbiota. Microorganisms that inhabit the human intestinal tract have a significant impact on our health: they participate in the functioning of the immune and endocrine systems, digestion, regulation of circadian rhythms, drug metabolism and other processes [1]. The composition of the microbiota undergoes a visible change from birth to death. The first years of a child’s life are important for the maturation of the microbiota [2], as this is the period of the “microbiota core” formation. Perhaps this is the period that is essential for maintaining health throughout life. With aging, gut microbiota acquires negative features: the number of opportunistic microorganisms grows, and the diversity of the microbiota decreases. Negative changes in microbiota are especially noticeable in patients with frailty syndrome, i.e., in patients with a negative aging scenario [3]. Due to a shift towards microorganisms that potentially may stimulate inflammation, the microbiota may contribute to the phenomenon of inflammaging [4], observed in an aging human body. Age-related changes in the microbiota appear to affect the intestinal permeability [5], which can be one of the components of the inflammatory vicious cycle in the elderly. For example, when transplanting a microbiota due to clostridial diarrhea, researchers observed not only an improvement in the microbiota picture, but also a restoration of normal barrier function and the levels of immune cells [6]. On the other hand, there are a small number of people who can be called “successfully aging,” which are long-livers (90 y.o. and older). As it turns out, the composition of their microbiota does not change so dramatically, and some studies suggest that the long-livers’ microbiome preserves high levels of beneficial bacteria and the potential for the synthesis of important metabolites against the background of a low representation of conditional pathogens [7,8]. In this study, we described the composition of the gut microbiota in a cohort of long-livers from Moscow, which has not yet been studied by using new generation sequencing.

## 2. Materials and Methods

### 2.1. Recruitment of Study Participants

The study included two groups of patients who signed their written consent to participate in the study. The first group of long-livers (LL) included 20 participants aged 97–100 years (4 men). The second group consisted of 22 participants aged 60–76 years (6 men) without clinical manifestations of chronic diseases (healthy elderly, HE). The cohort was described in the previous paper [9].

The LL group underwent a thorough physical examination with a comprehensive geriatric assessment, the Mini Mental State Examination (MMSE), nutrition analysis using the Mini Nutritional Assessment (MNA) scale, functional capacity with Instrumental Activities of Daily Living (IADL) and other tests, for blood pressure, heart rate and hand grip strength evaluation, and the assessment of geriatric syndromes. The LL also underwent a duplex scan of the carotid and femoral arteries as well as an expanded spectrum of biochemical analyses.

In the comparison group, a physical examination and an extended blood biochemical analysis were performed. All the subjects gave their informed consent for inclusion before they participated in the study. The study was conducted in accordance with the Declaration of Helsinki, and the protocol was approved by the Ethics Committee of The Stand-Alone Structural Unit of the Pirogov Russian National Research Medical University the “Russian Clinical Research Center for Gerontology” of the Ministry of Healthcare of the Russian Federation, Protocol #2, 18.03.2016.

### 2.2. Sample Collection and Gut Microbiota Analysis

All participants collected their feces in the sterile containers, and then placed it in a refrigerator. The samples were promptly delivered in an ice pack to the laboratory within not more than 5 h and then frozen at −20 °C. DNA was extracted from 125 mg of stool samples. Silica beads of 0.1 mm (300 mg) and 0.5 mm (100 mg), 125 μL of 4M guanidinium thiocyanate (pH 7.4), 260 μL of 10% N-lauroyl sarcosinate and 250 μL of 1× PBS were added to it. The mixture was vortexed and incubated at 70 °C for an hour. The cells were disrupted by bead-beating twice for 1 min with a 2 min interval on ice. Then, 15 mg of dry PVP was added to the lysate, and the homogenization process was repeated. The lysate was centrifuged at 12,000× *g* for 3 min at 4 °C. The supernatant was transferred to a new 2 mL tube and put it on ice. The equal volume of isopropyl alcohol was added and incubated at −20 °C for an hour and then centrifuged at 20,000× *g* for 15 min. The supernatant was removed, then 450 μL of 1× PBS was added to the pellet. The suspension was transferred to a new tube and resuspended. Then, 50 μL of 5M AcK was added. The mixture was put on ice for 90 min and centrifuged at 16,000× *g* for 30 min. The supernatant was transferred to a new tube, and 4 μL of RNAse was added (5 mg/mL). The mixture was incubated at 37 °C for 30 min. Then, 50 μL of 3M AcNa was added, followed by 1000 μL of 96% ethanol. The mixture was incubated at −20 °C for an hour and then centrifuged at 20,000× *g* for 15 min. The supernatant was removed and 300 μL of 70% ethanol was added, and the mixture was vortexed for 10 s and centrifuged for 5 min at 20,000× *g*. The supernatant was removed and 300 μL of 70% ethanol was added once again, and the mixture was vortexed for 10 s and then centrifuged for 5 min at 20,000× *g*. The supernatant was removed, and the pellet was incubated at room temperature with an open lid under an air draft for 15–20 min until it is completely dry. Then, 150 μL of TE buffer was added, and incubated at 4 °C for one hour. The DNA was stored at −20 °C.

Amplicon libraries of the V3–V4 region of the 16S rRNA gene were prepared using 16S Metagenomic Sequencing Library Preparation protocols. (https://support.illumina.com/content/dam/illuminamarketing/documents/products/appnotes/16S-Metagenomic-Library-Prep-Guide.pdf). Sequencing was performed at the MiSeq platform (Illumina, USA) according to the manufacturer’s recommendations.

### 2.3. Bioinformatics Analysis

Raw sequences were deposited in Sequence Read Archive (SRA) under PRJNA647648 accession number. The primary processing of the 16S rRNA gene sequencing data was carried out using the Knomics-Biota platform [10] in two different ways:

1- Using the closed-reference operational taxonomic unit (OTU) picking with Qiime 1.9 [11] and the GreenGenes [12] reference base with the preliminary cropping and filtering of low-quality reads (this analysis method is considered obsolete [13], however, it allowed us to compare the external data processed by this protocol as well as the HE Russian cohort);

2- By filtering the sequences using the DADA2 algorithm [14] with the variable trimming length (from 251 to 253 bp) followed by determining the taxonomy using QIIME2 [15] naive-bayes classifier trained on the GreenGenes database [12] (previously files with reads were rarefied to 20,000 reads per sample due to the computational complexity of the algorithm);

Statistical analysis was performed by using the Knomics-Biota platform [10] and R statistical programming language, version 3.3.0 (Core Team and Others 2013).

The first method of data processing was used for the comparison with publicly available metagenomic external data: Italy [16], Japan [17] and HE Russian cohort. The classified reads were rarefied to 5000 reads per sample. The comparison of the taxonomic composition between datasets was performed with the Wilcoxon rank-sum test (implemented in Knomics-Biota platform). Only taxa presented in at least 10% of samples at the level more than 0.2% were included in the analysis. Multiple testing adjustment was performed separately on each taxonomic level using the Benjamini–Hochberg procedure. The LEfSe (linear discriminant analysis (LDA) effect size) approach was used for effect size estimation [18].

The second method of data processing was used for the association discovery between the metadata and microbiome composition in the LL cohort. Associations between the bacteria abundance and metadata were discovered by applying a compositionality-aware algorithm—selbal [19]. The algorithm considers bacterial balances as predictors rather than abundance values themselves. The balance is defined as normalized log-ratio between the abundances of geometric means of two groups of bacteria: numerator and denominator groups. First of all, the optimal number of bacteria in the numerator and denominator groups were predicted. The best numerator and denominator members to predict the factor of interest are predicted on a whole dataset using a linear regression (balance~factor). Then, the stability of the balance members was assessed by 10 iterations of cross-validation procedure with ⅔ of samples for training and ⅓ for testing. The association was considered significant if the regression *p*-value was less than 0.05 after the multiple comparison correction (Bonferoni) and at the same time, the association was relatively stable (the bacteria was selected as the numerator or denominator in more than 50% of the cross-validation iterations). For alpha- and beta-diversity analysis, the classified reads were randomly rarefied to the 3000 reads per sample.

In each of two described processing protocols, alpha-diversity was estimated using the Shannon diversity metric, and beta-diversity by using the Bray–Curtis dissimilarity metric. The association of factors with alpha-diversity was assessed using Student’s *t*-test. The variance explained by factor in general microbiome composition was estimated using PERMANOVA and dbRDA analysis (adonis function in R).

## 3. Results

### 3.1. Comparison of the Microbiota Composition in Long-Livers and Conditionally Healthy Elderly

Long-livers’ microbiota composition turned out to be rather prosperous and significantly differed from the microbiota in elderly study participants (PERMANOVA, *R^2^* = 10.5%, *p* value = 0.001, Figure 1A). The analysis of taxa contribution to the main principal components shows that the gut microbiota of long-livers was more populated with symbiotic bacteria like unclassified genera from the *Ruminococcaceae* and *Christensenellaceae* family (Figure 1A). All taxa, significantly different between the two groups, are listed in Table 1.

Surprisingly, we found that such beneficial genera as *Lactobacillus, Christensenella, Roseburia* (Table 1), as well as the families *Ruminococcaceae, Christensenellaceae*, and *Lactobacillaceae*, were more represented in long-livers’ microbiome than in the healthy elderly.

Moreover, HE microbiota contained more bacteria that are considered to be conditional pathogens, including *Dialister* and *Peptostreptococcaceae, Dorea, Ruminococcus* (*Lachnospiraceae*). Thus, the composition of the LL microbiota looked quite healthy. Many beneficial bacteria were even more represented in their gut microbiome.

The diversity of the microbiota was higher in the group of elderly people (*p* = 1.7 × 10^−7^, Student’s *t*-test, Figure 1B). The mean Shannon alpha diversity index was 7.5 ± 0.5 and 6.39 ± 0.61 for the elderly people and long-livers, respectively.

### 3.2. Comparison of the Gut Microbiota of Long-Livers from Russia, Japan and Italy

We also analyzed whether the microbiota composition of our LL was similar to the microbiota of the LL from other countries. To this end, publicly available metagenomic data were used. The Russian LL group was compared with the LL from a Japanese study aged 91–104 [17], *n* = 24, and LL from Italy aged 99–109, *n* = 32 [16] and with younger inhabitants from the same countries. These countries are famous for their high life expectancy and a large number of centenarians.

When comparing the microbiota of LL from Russia and Japan, it turned out that the Moscow LL had a greater representation of *Akkermansia, Roseburia, Butyricimonas*, and some other bacteria (Table 2, Appendix A), while the Japanese LL had a greater representation of *Desulfovibrio, Fusobacterium, Peptostreptococcaceae*, which may largely depend on regional differences. Alpha diversity did not differ significantly (*p* = 0.8881, Student’s *t*-test). The differences were also reflected on the level of phylum: higher Proteobacteria and Fusobacteria abundance was observed for Japanese cohort, while Russian long-levers had higher *Firmicutes, Euryarchaeota* and *Verrucomicrobia* (Figure 3A). The mean Shannon alpha diversity index was 6.42 ± 0.97 and 6.39 ± 0.61 for Japan and Russian long-livers, respectively.

It is noteworthy that the LL microbiota composition from Russia was more similar to the Japanese LL than the microbiome of elderly people (60 to 80 y.o.) from the same study (*R*^2^ for the LL from Russia vs. Japanese comparison was 0.104, and the *R*^2^ for the Russian LL vs. Japanese elderly comparison was 0.131, Figure 3C,E).

When compared with the Italian study, we observed that the gut microbiota of the Russian LL was more similar to the Italian than the Japanese LL cohort (*R*^2^ for LL from Russia vs. Italy comparison was 0.053, and the *R*^2^ for LL from Russia vs. Japan comparison was 0.104, Figure 2). In Italian cohort, Russian LL microbiome was closer to Italian LL than to HE (*R*^2^ for LL from Russia vs. Italy comparison was 0.053, and the *R*^2^ for the Russian LL vs. Italy HE comparison was 0.065, Figure 2). Alpha-diversity in the Russian LL cohort (6.39 ± 0.61) did not differ with the microbiome diversity in the Italian LL cohort (6.29 ± 0.60, *p* = 0.5825, Student’s *t*-test).

When comparing the individual taxa abundance in the samples from Russian and Italian long-livers, a higher representation of *Coprococcus* and *Roseburia* was found in the Russian cohort. Interestingly, there was a lower abundance of *Desulfovibrio* in the Russian cohort (Table 3, Appendix A). The same trends were shown for the Japanese LL. At the phylum level, we observed that the Russian long-livers were distinguished by a higher Firmicutes abundance, as in the comparison with the Japanese cohort (Figure 2). Probably, the differences can be explained by the nutrition patterns.

### 3.3. Correlation between Gut Microbiota and Health Status

The sample size was quite small. However, we analyzed the microbiome relationship with the health features of long-livers. We used the selbal method to reduce the detection of false positive results. This method is described in the Materials and Methods section. We considered significant only bacteria which, in addition to being in the global balance, were also selected in more than 50% of cases for the balance according to the results of cross-validation (Figure 3).

We did not find any correlations between the microbiota composition and a list of parameters including cognitive status, functional capacity, hand grip strength, levels of vitamin D, microelements, proteins, lipoproteins and others. There probably could be more associations in a larger study. The values of the LL clinical parameters are shown in Table 4.

Some of the balances may seem to be unexpected. Thus, we observed that *Bifidobacteria* were positively associated with femoral arteries’ intima media thickness (IMT). Since we are considering a balance rather than a direct correlation, it can be assumed that not *Bifidobacteria*, but *Coprococcus* can indirectly reflect the state of the vascular wall. Moreover, the carotid artery IMT also negatively correlated with the level of *Coprococcus*, although the correlation did not reach significance (Table 5).

In addition, there were two associations beneficial to opportunistic bacteria balances. Balances with Bifidobacterium and *Faecalibacterium* in the numerator and *Coriobacteriaceae* in the denominator were found to be associated with the folic acid level and MNA score, respectively. In addition to the above, we also found a relationship between the diastolic blood pressure and the *Akkermansia/Blautia* balance.

Thus, the compositional data of the microbiome could be a reflection of some features of LL well-being.

## 4. Discussion

This work was the first study analyzing the gut microbiota composition among long-livers from Moscow by using new generation sequencing. Despite the fact that it has some limitations—first of all, a small sample size—interesting patterns and features were found.

The most surprising finding was the gut microbiota composition itself in these people. The microbiota had no negative traits. Conditional pathogens were poorly represented in comparison with relatively healthy elderly people of a younger age. While usually with aging, on the contrary, there is an increase in the number of opportunistic microorganisms [20]. Moreover, we observed that, in comparison with older people who did not have a severe pathology, long-livers had even higher levels of Lactobacilli and butyrate producers such as *Roseburia* and *Ruminococcus* from the *Ruminococcaceae* family. This is consistent with the hypothesis that microbiota can contribute to the development or inhibition of inflammaging. Perhaps with many bacteria synthesizing butyrate, gut microbiota may contribute to slower processes of inflammaging.

Biagi E. et al. recorded a higher level of Bifidobacterium in the group of long-livers older than 105 years compared with middle-aged people, however, this trend was observed in this group of supercentenarians, but not long-livers of a younger age [16]. It is noteworthy that, according to the same work, the presence of such positive bacteria such as *Akkermansia* and *Christensenellaceae* was also higher among long-livers. These results led the authors to conclude that maintaining a healthy microbiota can contribute to the longevity of study participants. Moreover, *Akkermansia*, which has a trend to be more presented in LL, may support normal gut permeability by enhancing tight junction functions [21].

Pathogens and conditional pathogens were poorly represented in the microbiota of long-livers. No differences with a younger cohort were revealed in these bacteria. Furthermore, the level of bacteria like *Peptostreptococcaceae* or *Betaproteobacteria*, the overrepresentation of which may have a negative effect, was significantly lower in the LL microbiota. For example, in a Japanese study [17] the level of the opportunistic bacteria was increased in the group of centenarians. The comparison with the centenarians from high life expectancy countries showed that the Russian LL microbiome had lower abundance of desulfating bacteria and a higher abundance of *Coprococcus* and *Roseburia*. We suppose that this should be due to the different dietary patterns between countries. Typical sulfate reducer *Desulfovibrio* could reflect a higher consumption of seafood by Italians and Japanese [22], while *Roseburia* could be a sign of a large amount of cereal grains in the Russian diet [23]. In addition, Russian LL microbiome had more similarities with the Italian cohort, which is geographically closer.

Finally, our findings in the correlation of clinical features with microbiota composition were quite modest. Probably in a larger study, we could see more associations of clinical status with microbiota. Nonetheless, adequate nutrition was associated with beneficial to opportunistic bacteria balance. A higher folic acid level was positively correlated with Bifidobacterium, the MNA score with *Faecalibacterium* and both were negatively associated with *Coriobacteriaceae*. Notably, folic acid is produced by Bifidobacterium in the gut [24] and could be absorbed in the colon [25]. Faecalibacterium is one of the main butyrate producers with anti-inflammatory properties. Faecalibacterium is sensitive to a healthy diet with high fiber consumption [26]. However, again, we should consider these bacteria not as in direct relationships, but namely in balance with *Coriobacteriaceae*. According to the selbal analysis, the Bifidobacterium to *Coprococcus* and *Faecalibacterium* to *Coprococcus* balances were associated with femoral and carotid IMT (the latter not significantly). Notably, few studies have shown an increase in bifidobacteria in some pathologies, such as Parkinson’s disease [27,28]. Nevertheless, we should consider this result as a balancer of bacteria, and not as a direct correlation. It can be assumed that these balances may somehow reflect vascular health, although this hypothesis requires larger studies.

## 5. Conclusions

The results of this study show that the gut microbiota of long-livers is maintained in a fairly good condition. We suggest that microbiota may play an important role in longevity. Probably, high levels of butyrate producing and other beneficial bacteria, which potentially reduce systemic low-grade inflammation, prevents the early development of age-related diseases. Nevertheless, the present study is rather small for unambiguous conclusions and therefore further extensive longitudinal studies are needed for a deeper understanding of the microbiota influence on life expectancy.

## Figures and Tables

**Figure 1 microorganisms-08-01162-f001:**
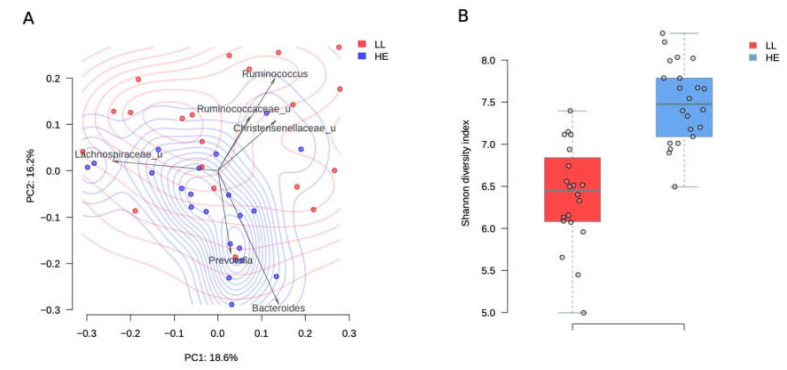
Comparison between long-livers and healthy adults from a Russian population. A-PcoA (Principal Coordinates Analysis) of Russian long-livers (blue dots) and healthy elderly participants (red dots) samples using the Bray–Curtis dissimilarity metric. Arrows show the top taxa in terms of the explained variance in given axes. The arrows’ length is proportional to the percent of variance explained by the taxon. The arrows’ angle reflects the distribution of this variance between the axes. B-Gut microbiota Shannon alpha-diversity indices in long-livers (**A**) and conditionally healthy elderly participants (**B**).

**Figure 2 microorganisms-08-01162-f002:**
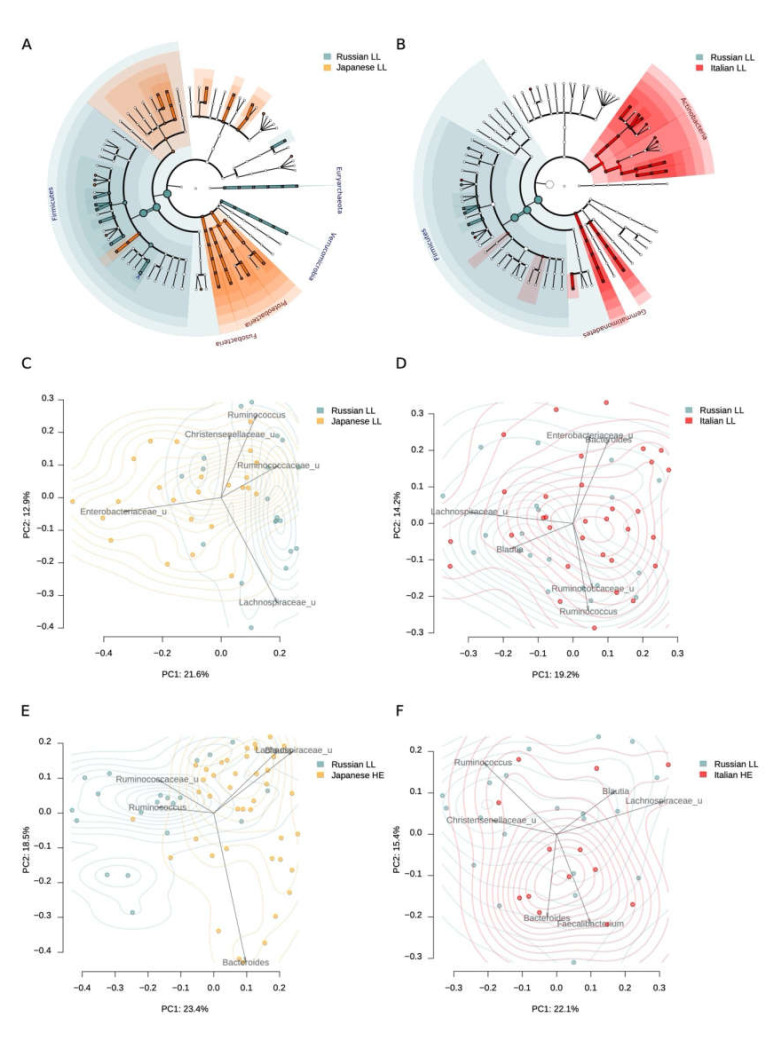
Gut microbiome differences between the long-livers from Russia and the long-livers and healthy adults from external datasets: Italian and Japanese. A, B—Generalized visualization of the differences in the individual taxa abundance at different taxonomic levels between the Russian and Japanese long-livers (**A**), and the Russian and Italian long-livers (**B**). Significant differences in the phylum level are marked with labels. C, D—Comparison of the long-livers’ microbiome community from Russian (*n* = 20) and Japanese (*n* = 24) cohorts (**C**) and the Russian (*n* = 20) and Italian (*n* = 32) cohorts (**D**) using PcoA analysis with Bray–Curtis dissimilarity metric. E, F—Comparison of Russian long-livers’ (*n* = 20) microbiome community with that of Japanese (*n* = 24) (**E**) and Italian (**F**) (*n* = 32) healthy adults.

**Figure 3 microorganisms-08-01162-f003:**
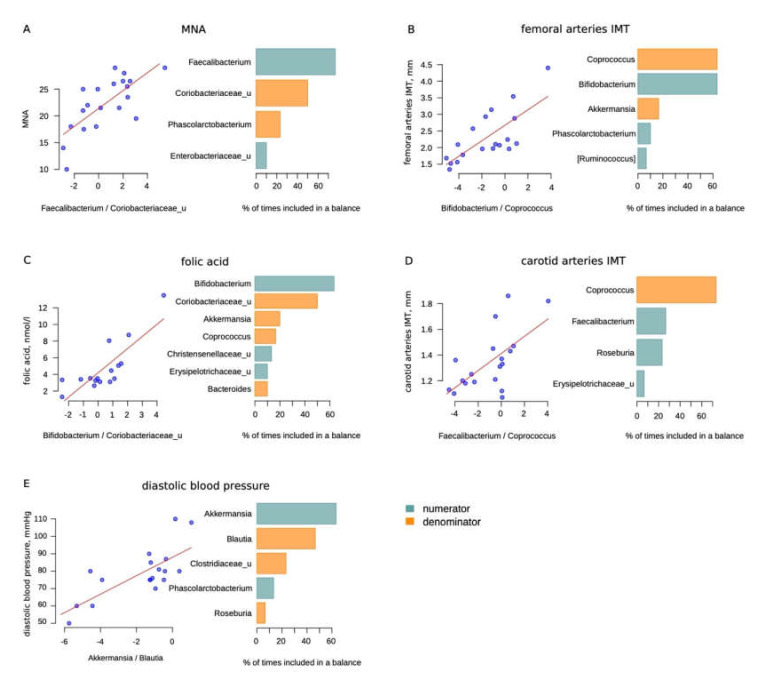
Selbal analysis results: (**A**)—Mini Nutritional Assessment (MNA) scale; (**B**)—femoral arteries intima media thickness (IMT), mm; (**C**)—folic acid, nmol/L; (**D**)—carotid arteries IMT (not significant), mm; (**E**)—diastolic blood pressure, mmHg (less than 50% of cases for balance according to the results of cross-validation for Blautia). The scatter plots on the left illustrate the regression between the balances determined on the whole dataset and the analyzed factor. The barplots on the right illustrate the stability of the three most frequent balanced members during the cross-validation procedure: the blue bars correspond to the numerator members and orange—to the denominator members.

**Table 1 microorganisms-08-01162-t001:** Bacteria with significantly different abundance in the LL and HE groups.

Taxon	Taxa Level	LDA Score	*p*-Value	Adjusted *p*-Value	Group with Higher Abundance
*f__Ruminococcaceae*	family	4.6572	0.001	0.006	LL
*f__Christensenellaceae*	family	4.2216	0.001	0.006	LL
*f__Lactobacillaceae*	family	3.7116	0.001	0.006	LL
*g__u(f__Ruminococcaceae)*	genus	4.3331	0.006	0.030	LL
*g__u(f__Christensenellaceae)*	genus	4.2426	0.001	0.007	LL
*g__Roseburia*	genus	4.0216	<0.0001	<0.0001	LL
*g__Lactobacillus*	genus	3.9426	<0.0001	<0.0001	LL
*c__Betaproteobacteria*	class	3.5691	<0.0001	0.003	HE
*o__Burkholderiales*	order	3.5490	<0.0001	0.003	HE
*f__u(o__Clostridiales)*	family	4.5760	<0.0001	<0.0001	HE
*f__Veillonellaceae*	family	3.7684	0.002	0.008	HE
*f__(Mogibacteriaceae)*	family	3.5914	0.001	0.006	HE
*f__Alcaligenaceae*	family	3.5608	0.001	0.006	HE
*f__Peptococcaceae*	family	3.3366	0.004	0.015	HE
*f__Peptostreptococcaceae*	family	3.1999	0.011	0.037	HE
*g__u(o__Clostridiales)*	genus	4.5985	<0.0001	<0.0001	HE
*g__Dorea*	genus	4.2574	<0.0001	0.001	HE
*g__Sutterella*	genus	4.0443	0.001	0.007	HE
*g__u(f__Peptostreptococcaceae)*	genus	4.0241	0.004	0.021	HE
*g__(Ruminococcus) (Lachnospiraceae)*	genus	3.9725	0.001	0.008	HE
*g__Dialister*	genus	3.7867	<0.0001	0.004	HE

Note: here and later: g—genus, f—family, p—phylim, u—unit.

**Table 2 microorganisms-08-01162-t002:** Differences in the microbiota composition between the LL from Japan and Russia at the genus level (Wilcoxon rank-sum test).

Taxon	More Presented in	*p*-Value	Adj. *p*-Value	LDA Score
*g__u(f__Enterobacteriaceae)*	Japanese LL	0.004	0.018	4.461
*g__Enterococcus*	Japanese LL	<0.0001	0.004	4.321
*g__Parabacteroides*	Japanese LL	0.003	0.013	4.128
*g__u(f__Rikenellaceae)*	Japanese LL	0.006	0.023	3.913
*g__Butyricimonas*	Japanese LL	0.017	0.047	3.887
*g__Granulicatella*	Japanese LL	0.005	0.021	3.835
*g__Fusobacterium*	Japanese LL	<0.0001	0.001	3.827
*g__u(f__Peptostreptococcaceae)*	Japanese LL	0.001	0.004	3.747
*g__Desulfovibrio*	Japanese LL	<0.0001	0.001	3.744
*g__Sutterella*	Japanese LL	0.009	0.031	3.670
*g__u(f__Ruminococcaceae)*	Russian LL	<0.0001	0.001	4.591
*g__u(f__Lachnospiraceae)*	Russian LL	0.006	0.023	4.508
*g__Akkermansia*	Russian LL	0.011	0.033	4.291
*g__Coprococcus*	Russian LL	0.001	0.006	4.274
*g__Dorea*	Russian LL	<0.0001	0.001	4.033
*g__Methanobrevibacter*	Russian LL	<0.0001	0.000	3.863
*g__Roseburia*	Russian LL	<0.0001	0.003	3.787
*g__u(f__Coriobacteriaceae)*	Russian LL	0.011	0.033	3.536

**Table 3 microorganisms-08-01162-t003:** Comparison of the LL microbiota composition of the centenarians from Russia and Italy at the genus level (Wilcoxon rank-sum test).

Table.	More Presented in	*p*-Value	Adjusted *p*-Value	LDA Score
*g__Coprococcus*	Russian LL	0.006	0.042	4.195
*g__Dorea*	Russian LL	0.000	0.000	4.003
*g__Roseburia*	Russian LL	0.004	0.033	3.467
*g__Eggerthella*	Italian LL	0.000	0.000	3.446
*g__u(f__Coriobacteriaceae)*	Italian LL	0.007	0.044	3.415
*g__Coprobacillus*	Italian LL	0.000	0.000	3.362
*g__u(c__Gemm-1)*	Italian LL	0.000	0.000	3.251
*g__Desulfovibrio*	Italian LL	0.008	0.046	3.218
*g__Nesterenkonia*	Italian LL	0.000	0.000	3.215
*g__Actinomyces*	Italian LL	0.002	0.017	3.192

**Table 4 microorganisms-08-01162-t004:** Clinical characteristics of the LL group.

Factor	Median	IQR
Body mass index, kg/m^2^	25.10	5.66
Local frailty scale (0–7)	3.00	1.25
Systolic blood pressure, mmHg	155.00	32.50
Diastolic blood pressure, mmHg	78.00	9.00
Heart rate, per minute	69.00	9.00
Geriatric depression scale	6.00	7.25
IADL	16.00	9.25
MNA	22.75	7.00
Maximum carotid stenosis, %	50.00	7.50
Carotid IMT, mm	1.31	0.25
Femoral IMT, mm	2.09	0.85
Glycated hemoglobin, %	5.79	0.50
Protein, g/L	67.40	2.85
Creatinine, mg/dL	89.40	22.23
Mg, mmol/L	0.88	0.11
Fe, μmol/L	12.90	5.40
C-reactive protein, mg/L	2.06	3.91
Folic acid, nmol/L	3.50	1.91
Ionized calcium, mmol/L	1.05	0.05
Vitamin B12, pg/mL	261.00	125.00
NT-proBNP (n terminal fragment in the prohormone of brain natriuretic peptide), pg/mL	976.30	1793.88
Triglycerides, mmol/L	1.04	0.34
High density lipoproteins, mmol/L	1.43	0.48
Low density lipoproteins, mmol/L	3.55	1.18
Atherogenic index	2.89	1.31
Grip strength, kg	17.00	6.38
Montreal Cognitive Assessment	11.50	18.00
MMSE	23.00	25.00

**Table 5 microorganisms-08-01162-t005:** Selbal analysis results: the correlation between the gut microbiota composition and the health status.

Factor	Association Direction	Bacteria	*R* ^2^	*p*-Value for the Appropriate Balance	Adjusted *p*-Value for the Appropriate Balance
Femoral arteries IMT	+	*Bifidobacterium*	0.5425	0.0003	0.0086
	-	*Coprococcus*			
Carotid arteries IMT	+	*Faecalibacterium*	0.3868	0.0044	0.0876
	-	*Coprococcus*			
Folic acid	+	*Bifidobacterium*	0.6486	0.0001	0.0028
	-	*Coriobacteriaceae*			
MNA	+	*Faecalibacterium*	0.4941	0.0003	0.0086
	-	*Coriobacteriaceae_u*			
Diastolic blood pressure	+	*Akkermansia*	0.5218	0.0004	0.0108
	-	*Blautia **			

Note: *—less than 50% of cases for balance according to the results of cross-validation.

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
