# Peer review of "A Cross-Sectional Study of the Gut Microbiota Composition in Moscow Long-Livers"

_microorganisms, 2020, doi:10.3390/microorganisms8081162_

Round 1
Reviewer 1 Report
The present study analysed the long-livers microbiota of Moscow’s population in comparison to elderly, and in comparison to Italian and Japanese cohorts of already published studies. Although the study has limitations that need to be listed/resolved, it is interesting and it deserves to be published. First of all a careful English revision is necessary. Moreover the materials and methods need a strong rewrite, especially for the section 2.2. The data presentation in the figures should be improve, also with more detailed description in the captions. Finally, the results should be better explained with clearer references to the data presented. Some more suggestions are shown below.
Lines 18-19: please explain better as ‘4 of whom were men’
Lines 67-68: because the cohort is different for the presence of long-livers, please describe here the conditions.
Line 76: please provide the approval number of the study and the institution of the ethics committee.
2.2 section: please indicate the mean time from collection to frozen. Please sequentially describe the methods from DNA isolation to sequencing (lines 80-82). Lines 83-84 the verbal form is missing. Please describe the homogenization methodology (instruments, and so on).
Fig.1, 5, 6: in my opinion the data presentation is not useful to understand what the authors stated
Table 1 and 2: in my opinion the combination of the two tables in one single table can help the reader to better understand the results.
Figure 2: the colored boxes are useless if you put the name also on the axis.
Figure 3: please improve the quality and the information of this figure.
Lines 192-193: the comparison of different number statistical population is incorrect, and it should be indicated as a limitation of the study. The same is for the different ages of the studies’ participants.
Please sequentially describe the results and the figures, I mean fig.6 should be 5.
Author Response
Dear Reviewer,
Thank you for careful and thorough reading of this manuscript and for your valuable comments. Our response follows.
Lines 18-19: The correction has been made.
Lines 67-68: The cohort was previously described in the paper «Kashtanova, D.A.; Tkacheva, O.N.; Doudinskaya, E.N.; Strazhesko, I.D.; Kotovskaya, Y.V.; Popenko, A.S.; Tyakht, A.V.; Alexeev, D.G. Gut microbiota in patients with different metabolic statuses: Moscow study. Microorganisms 2018, 6.». In order not to duplicate the information, we’ve left a link to this article in the paper.
Line 76: The correction has been made.
2.2 section: The correction has been made.
Fig.1, 5, 6: The corrections have been made.
Table 1 and 2: We’ve combined the tables.
Figure 2: We added contour information on PcoA plot to highlight the visual differences between classes. For Figure 1 the text (“On the graph of multidimensional scaling, we can see that the gut microbiota of long-livers was more populated with symbiotic bacteria like unclassified genera from the Ruminococcaceae and Christensenellaceae family”) is supported by the arrows, which shows the top taxa in terms of explained variance in two first PCs. We've added a description of arrows meaning in the Figure legend. We also combined Figure 1 and Figure 2 and removed x axis labels from Figure 2.
Figure 3 and Lines 192-193: Actually, when we’ve made the conclusions, mentioned in the text, we relied rather on R2 values than on visual analysis. For this reason we’ve moved R2 values from the Figures to the text. However, we think that the effect is reflected on the plots as well. To facilitate visual analysis we added contours to the plots. We combined Figures 3, 4 and 6 so one can easier compare them. We removed Figure 5 because the information on the Figure is duplicated by Figures 5 and 6. We also decided to add cladogram to Italian LL vs Russian LL comparison to harmonize results presentation for Japanese and Italian cohorts.
Thank you once again for your constructive suggestions, which help to improve the quality of this manuscript.

Reviewer 2 Report
Dear Authors,
Here my comments,
1. Abstract: To avoid the abbreviations, specially if you have not been used them previously.
Please re-write according to.
2. Figures. Please, expand the "box-point" of the legend, so the space between 2 closed legends.
3. Figures. Please, change the orientation of x-axes or horizontal axes
4. Tables. Define o", "u", "f" and "g"
5. An improvement in the results´exposition or explanation is necessary.
6. Improve the english style.
Author Response
Dear Reviewer,
Thank you very much for reading and working with our manuscript and for your valuable comments. Our response follows.
Abstract: The corrections have been made.
Figures: We’ve expanded the "box-point" of the legend and changed the orientation of x-axes.
Tables: We've made a note with the decoding of these values.
5-6: The corrections have been made.
Thank you once again for the constructive comments and help with improving the manuscript!

Round 2
Reviewer 1 Report
The authors improved their manuscript making the suggested changes.